# How Different Predominant SARS-CoV-2 Variants of Concern Affected Clinical Patterns and Performances of Infected Professional Players during Two Soccer Seasons: An Observational Study from Split, Croatia

**DOI:** 10.3390/ijerph20031950

**Published:** 2023-01-20

**Authors:** Jasna Nincevic, Anamarija Jurcev-Savicevic, Sime Versic, Toni Modric, Ante Turic, Ante Bandalovic, Boris Becir, Marijana Mijakovic, Ivana Bocina, Damir Sekulic

**Affiliations:** 1Teaching Public Health Institute of Split and Dalmatia County, 21000 Split, Croatia; 2School of Medicine, University of Split, 21000 Split, Croatia; 3Department of Health Studies, University of Split, 21000 Split, Croatia; 4Faculty of Kinesiology, University of Split, 21000 Split, Croatia; 5HNK Hajduk Split, 21000 Split, Croatia; 6Department of Orthopedics and Traumatology, University Hospital Split, 21000 Split, Croatia; 7High Performance Sport Center, Croatian Olympic Committee, 10000 Zagreb, Croatia

**Keywords:** COVID-19, variant of concern, clinical presentation, clustering, preventive measures, football, performance, global positioning system

## Abstract

There are limited data describing clinical patterns and match running performance (MRP) among players with COVID-19 infection before and after infection, particularly related to different predominant SARS-CoV-2 variants, as well as in comparison to uninfected players. This observational study was conducted during two consecutive soccer seasons in one professional club in Split, Croatia. There were four clusters of mild, self-limited, or asymptomatic infection characterised by low adherence to preventive measures. Infected players had significantly more symptoms (*t*-test = 3.24; *p* = 0.002), a longer period of physical inactivity (χ^2^ = 10.000; *p* = 0.006) and a longer period of self-assessment for achieving full fitness (χ^2^ = 6.744; *p* = 0.034) in the 2020–2021 season (Wuhan wild strain and Alpha variant) than in the 2021–2022 season (Omicron variant). It was also found that, despite the milder clinical presentation of the infection in the 2021–2022 season, the players had significantly more abnormal laboratory findings (χ^2^ = 9.069240; *p* = 0.002), although without clinical significance at the time of the study. As for the MRP, player performance in the 2021–2022 season was not negatively affected by the Omicron variant, while there was an improvement in MRP in scores for a sample of all players. The RTP protocol was correctly applied because it helped the athletes to recover their pre-infection physical capacities relatively quickly. This study advances the understanding that an optimally and individually planned RTP protocol is crucial for the MRP of infected players. Future research needs to replicate the findings of abnormal laboratory results and extend the study focusing on their potential long-term clinical significance.

## 1. Introduction

For the last three years the world has been changing dramatically due to the COVID-19 pandemic. The emergence of this unique pathogen has presented one of the most serious global health threats in the 21st century which humanity has faced [1]. A variety of control measures, especially lockdowns on different levels, stopped or limited many aspects of social life, including sport. This pandemic profoundly influenced sport as one of the generators of social and economic development, regardless of its level: professional, amateur, or recreational [2]. The heavy burden of this pandemic on professional sport was manifested through the restriction of opportunities for training as well as cancellation or postponement of sport events with serious financial loss [3]. It seems to be clear that one cannot avoid the virus throughout life, but that its impact can be minimised, even in sports.

With time, professional sport has continued, with specific anti-epidemic measures applied. As soon as the control measures were relaxed, the first clusters of COVID-19 infection among players in sports clubs began to appear [4]. Professional athletes continued to travel within the country and abroad due to training and competitions, while permitted sports events continuously posed a risk of disease, mainly due to the difficulties in maintaining physical distance [5].

Similar to the general population, the clinical presentation of COVID-19 among professional athletes has been associated with a wide range of symptoms, mainly respiratory and systemic symptoms [6].

Furthermore, COVID-19 can affect other organ systems, including the cardiovascular, neurological, gastrointestinal, and renal systems, as well as skeletal muscle function [7]. However, even mild/moderate clinical presentation of COVID-19 can be associated with prolonged or “Long-COVID” symptoms in previously perfectly healthy individuals [8]. Although the majority of professional athletes recovered from a mild form of COVID-19 without complications or the need for hospitalisation, a smaller number of them reported persistent/residual symptoms several weeks to months after contracting the disease. These symptoms included coughing, tachycardia, and extreme exhaustion. They were mostly reported by athletes with lower respiratory system infection [9,10,11,12].

Three main challenges were highlighted as related to COVID-19 among professional athletes, according to Hull: detection of serious complication such as heart disease in a timely manner, safely graduated return-to-play (RTP), and detection and management of persistent symptoms [13]. Cardiac complications such as myocarditis/pericarditis among athletes, especially when unrecognised, may pose a health risk, particularly when they return to a high level of exercise and physical activity. Therefore, the key role of sports medicine is to provide guidelines for the safe RTP after recovering from COVID-19 [14]. A number of publications presented the recommendations on RTP protocol following COVID-19 in professional athletes [4,15,16].

In one study focused on match running performance (MRP) of professional soccer players during the 2020/21 season, poorer performances regarding high-intensity acceleration and high-intensity decelerations in infected players prior to and post COVID-19 infection were found. Among the COVID-19-positive soccer players, a smaller part (6%) had a longer period of adaptation to usual training routines due to poor physical condition, although their medical parameters were satisfactory [4]. In order to minimise the impact of lack of training during isolation, some athletes were encouraged to perform adapted training sessions at home [17].

In the Republic of Croatia, the first case of COVID-19 was recorded on 25 February 2020, which was followed by the implementation of general and specific anti-epidemic measures at the national and regional level, in response to the current epidemiological situation. By the decision of the Civil Protection Headquarters of the Republic of Croatia to prevent the spread of the coronavirus from 19 March 2020, among other things, all sports competitions and organised trainings as well as the gyms, sports, fitness, and recreation centres were closed. The last official match of the Croatian football competition took place on 9 March 2020 [18]. Over time, the control measures became more relaxed and sport activities were allowed outdoors, and then also indoors. All control measures were developed by the Croatian Public Health Institute and changed over time [19].

It has been witnessed that SARS-CoV-2 is highly prone to mutations. Many SARS-CoV-2 variants have been detected from the first days in January 2020. However, only a limited number of variants have had substantial biological significance in terms of increased transmissibility, disease severity, and ability to escape vaccine-induced and natural immunity. These variants have been named from a public health perspective as “variants of concern” (VOC) [20].

The European Centre for Disease Prevention and Control (ECDC) recommends tracking the relative proportion of the SARS-CoV-2 variants over time and suggests how to estimate and collect a representative sample size. Since 9 February 2021, Croatia has monitored novel or emerging viral variants by whole-genome sequencing according to the ECDC protocol [21]. Representative and targeted positive samples have been sequenced weekly and the predominant virus variants in circulation in Croatia have been documented, with the data broken down by county [22,23].

In contrast to numerous studies that analysed the symptoms and course of the disease in the general population [6,7], there are a limited number of studies reporting patterns of COVID-19 infection among professional athletes and comparing their MRP before and after infection [24]. To the authors knowledge, no study has thus far compared clinical characteristics and performance among professional athletes associated with different dominant VOCs. It is also worth noting that the studies usually described the burden of COVID-19 among players, but lacked data related to transmission chain and cluster emergence. This is of practical value regarding implementation of control measures, their compliance and supervision over their implementation, especially when this pandemic is not over yet, and other emergent or re-emergent respiratory pathogens may have shared the same epidemiological pathways.

This observational study aimed to evaluate (i) clinical characteristics of COVID-19 infections among a group of professional soccer players with a positive PCR test for SARS-CoV-2 during two consecutive Croatian professional soccer seasons with different prevailing VOCs such as the dynamics of infection, main symptoms, and severity of disease, resting period during illness, RTP, persistent symptoms, as well as abnormal laboratory and cardiac test results after recovery, (ii) chain of transmission causing clusters of infection (iii) the differences of match running performances among infected players before and after infection; among infected players compared to uninfected; as well as between the two soccer seasons.

## 2. Materials and Methods

### 2.1. Study Design and Participants 

This observational study includes two whole populations of professional male soccer players from the first division of one soccer club in Split, the second biggest city in Croatia. This club participates in the first national soccer league along with nine other soccer clubs. 

One population consisted of all players (N 47) in season 2020/2021 (season 1) and the other consisted of all players (N 31) in season 2021/2022 (season 2). 

Season 1 started on 13 August 2020 and lasted until 22 May 2021, while season 2 started on 17 July 2021 and lasted until 21 May 2022. 

This study used two sets of variables; variables related to COVID-19 and match running performances.

The participants in each season were divided in two subgroups based on the results of PCR test for SARS-CoV-2 positive (infected-INF) and negative players (noninfected-NONINF). The definition of an infected-INF (COVID-19 case) player was considered as a person who met laboratory criteria for a confirmed case (detection of severe acute respiratory syndrome coronavirus 2 ribonucleic acid in a clinical specimen using a molecular amplification detection test) regardless of the presence of symptoms [25]. 

This research relied on a regular rapid intervention of the epidemiological service regarding the appearance of the first cluster of COVID-19 among the players. The definition of a cluster of COVID-infection was set as two or more laboratory-confirmed COVID-19 cases among players with onset of illness or if asymptomatic, with a positive test result within a 14-day period, who are epidemiologically linked (have a potential connection in time and place) within the soccer club [26].

In both seasons, all nasopharyngeal and pharyngeal swabs were tested using real-time reverse transcriptase polymerase chain reaction (RT-PCR) for the detection of SARS-CoV-2 at certified laboratories by Croatian Health Insurance Fund. These swabs were collected by medical staff from official laboratories. Test results were available within 24 h in most cases. According to regulations of the Croatian national soccer federation, there was no obligation to test during the season or before matches. The players were tested following the presence of symptoms and during a cluster investigation when all the team was tested. The testing was initiated by the team physician or epidemiologist and it was not repetitive for non-infected players unless the presence of the symptoms occurred. 

According to the Croatian quarantine rules which changed over time, all individuals who tested positive for COVID-19 have to be isolated for 14 days (later 10 days) regardless of their clinical symptoms. After isolation, they underwent RTP protocols before starting full team training again.

All COVID-19 positive players were interviewed by an epidemiologist assigned to the club by the Teaching Public Health Institute of Split and Dalmatia County and who followed the club throughout the pandemic.

The epidemiologist interviewed players using standard data collection for COVID-19 case reporting in Croatia in the general population [27] which was changed over time with additional questions regarding vaccination and persistent symptoms. The interview was done immediately after a positive test, and then again one month after testing. Some extra questions were added regarding the number of days to return to full form after the termination of isolation and the number of days with no physical activity, during and after the isolation caused by a COVID-19 infection. These questions were pilot tested for clarity and consistency on several participants and modified accordingly.

After isolation, the participants were subjected to a physical examination by team physicians and a cardiologist. A set of laboratory parameters was analysed: (i) complete blood count with differential tests, (ii) comprehensive metabolic panel (glucose, creatinine, urea, urate, bilirubin, aspartate aminotransferase, alanine aminotransferase, gamma-glutamyl transferase, alkaline phosphatase, sodium, potassium, calcium, iron, total iron binding capacity, unsaturated iron binding capacity), (iii) lipid panel (cholesterol, triglycerides high-density lipoprotein, low-density lipoprotein), (iv) urine test strips.

In season 2, thromboinflammatory biomarkers such as C-reactive protein (CRP), D-Dimer, high sensitivity troponin (hs-Tn), and marker of heart failure N-terminal proBrain Natriuremic Peptide (NT- proBNP) were analysed. 

Cardiological evaluation included a cardiologist examination followed by a 12-lead electrocardiogram, ergometry, and heart ultrasound while a chest X-ray was performed upon indication. 

In this study data was used regarding dominant virus variants isolated at a national level by genome sequencing. Strategy and sample selection for representative sampling of SARS-CoV-2 cases for genomic monitoring was carried out by the Croatian Public Health Institute according to the European Centre for Disease Control and Prevention (ECDC) guidelines for monitoring the level of circulation of variants at the national level [21]. Some of the samples of the participants were included in the genome sequencing at the national level. The results of the sequencing of SARS-CoV-2 samples were reported to ECDC and entered into the international database (Global Initiative on Sharing Avian Influenza Data—GISAID). The sequencing results can be seen using a visualisation tool that presents the results obtained from the analysis of the entire genome of SARS-CoV-2 from the Republic of Croatia [28].

The dominant virus variants in Season 1 were Wuhan wild strain and Alpha VOC, while Omicron VOC was dominant in the time of cluster in Season 2 [28].

In this club, MRP was routinely measured. For the purpose of this study, the MRP measurement of players who had professional status on the team during both seasons and who have actively participated in soccer for at least 10 years were included. 

MRP performance results in Season 1 were already published. The same protocol for inclusion and exclusion criteria were followed for MRP performances in season 1 [2]. In brief, INF players included in the study had a positive SARS-CoV-1 PCR test, participated in at least one game (minimum of 60 min) during the period one months before COVID-19 diagnosis, and participated in at least one game (minimum of 60 min) played at least one month after their return to play after COVID-19 isolation. NONINF players included in the study were not diagnosed for COVID-19, had no other disease or condition which prevented them from training or competing for more than 10 days (20 days cumulatively over the half-season) and participated for 60 min in at least two games during the first half of the season.

The goalkeeping playing position was an additional exclusion criterion only for MRP, regardless of their results of a PCR test for SARS-CoV-2. 

The same MRP measurements which were performed during the 2020/2021 soccer season in Croatia were repeated in the same professional soccer team. The detailed protocol was described by these authors in another reference [4]. 

In brief, GPS devices were used with a sampling frequency of 10 Hz [Vector S7, Catapult, Catapult Sports Ltd., Melbourne, Australia]. Validity and reliability of these devices have already been confirmed [29].The following MRP parameters were included: the total distance covered (m]; distance covered at different speeds—low-intensity running (<14.3 km/h), running (14.4–19.7 km/h), high-intensity running (>19.8 km/h), high-speed running (19.8–25.1 km/h), and sprinting (>25.2 km/h); total accelerations and decelerations (>±0.5m/s^2^); and accelerations and decelerations performed at high intensity (>±3 m/s^2^). For players infected with SARS-CoV-2, MRPs were observed prior to and post infection, and the inclusion criterion for the games was that they occurred one month before the infection (pre) and one month after end of the isolation period (post). Since all players were infected at the same time, the same sample of the games was used for the noninfected players. The suggestion of Stevens et al. to evaluate at least 60 min per game was accepted in the approach [30].

Epidemiological data for all players infected with SARS-CoV-2 was collected, while MRPs were measured only for those players who fellfield criteria for participation (13 players). 

### 2.2. Statistical Analysis

Descriptive statistical methods were used, namely frequencies and means, to describe the dataset. The data from two seasons was considered to be two independent datasets. The majority of players were different in the two seasons. It is known that immunity after recovery from infection is not permanent or lifelong, so decision was taken to disregard the impact of recovering from it during season 2. An χ^2^ test of independence was used to determine if the number of infected players, the number of reinfections, the number of asymptomatic players, the number of previously infected players, and the presentation of abnormal laboratory findings depended on the season. A comparison of means two-tailed *t*-test with the assumption of unequal variances was used to determine the difference in the average number of reported symptoms per player among seasons; the difference in average number of reported symptoms per player between first time infected players and reinfected ones; the self-assessed number of days to return to full form average between the group with the fever as a reported symptom and the group without it; and the self- assessed number of days to return to full form average between the players in season 2 who were infected for the first time and those who were reinfected. The number of days with no physical activity were divided into 3 groups; namely 0–4 days, 5–9 days, and 10 plus days. In order to determine if there was any seasonal dependency an χ^2^ test of independence was applied. Simple linear regression was used to test if the number of symptoms significantly predicted the number of days with no physical activity. The self-assessed number of days to return to full form was divided in 3 groups; namely 0–7 days, 8–14 days, and 15 and more days. In order to determine if there was any seasonal dependency an χ^2^ test of independence was used. Simple linear regression was employed to test if the number of symptoms significantly predicted the self-assessed number of days to return to full form. R^2^ with the linear regression data and Cohen’s d with the comparison of means was used as a measure of the magnitude of the effect in the reporting. An χ^2^ online calculator was used at https://www.statology.org/chi-square-test-of-independence-calculator (accessed on 10 December 2022). Comparison of means *t*-test and linear regression were calculated by Microsoft Office Standard 2013 Excel software package.

Running variables were descriptively analysed (arithmetic mean and standard deviations). To establish possible differences pre- and post- COVID-19 infection and between the groups of infected and non-infected players, repeated measures ANOVA was used. 

A *p* value of less than 0.05 was accepted as indicating statistical significance.

## 3. Results

All players in both seasons were included in the study. There was no refusal of testing after a doctor’s recommendation or to MRP measurement.

During season 1, 66% (31 of 47) players were tested positive for SARS-CoV-2, while in season 2 51.6% (16 of 31) which was not of statistical significance (χ^2^ = 1.604, *p* = 0.205). In season 1 one player was infected before the start of soccer season in May 2020, then re-infected in February 2021, while in season 2 a total of 15 players were identified as being previously infected (χ^2^ = 24.516, *p* = 0.000001). In season 2, 6 out of 15 previously infected players were re-infected in comparison to just 1 in season 1 (χ^2^ = 9.780, *p* = 0.002) (Table 1).

Three players, infected during season 1, were sporadic cases with no further spreading, while three clusters of infection were observed during winter. The first cluster was notified from 5 to 11 November 2020 with eight positive players, the second from 30 November to 8 December 2020 with five positive players, while the third cluster, in February/March 2021, had a total of fifteen infected players (Figure 1). The causative agent of the first and second cluster was probably the Wuhan strain of SARS-CoV-2, while the third cluster was caused by SARS-CoV-2 variant Alpha B1.1.7 which were dominant at that time at national level. The samples of positive players from the third cluster were included in whole-genome sequencing as a part of virus variants surveillance at national level. In season 2, one cluster with 16 positive players was notified from 23 December 2021 to 5 January 2022. The samples were included in genome sequencing and in this cluster SARS-CoV-2 variant Omicron B1.1.529 was detected.

All clusters from the first season occurred after away games when the players travelled together by bus. The appearance of the first and the second clusters (October and November 2020) was related to one-day trips. During travel to the match, one of the players was in the incubation phase, without symptoms. On the way back home, the symptoms appeared and he infected the players who were sitting closest to him. During epidemiological investigation it was found that during the trip the epidemiology recommendations for preventive measures were not strictly followed, as the players would occasionally take off their masks and consume food or drink.

The third cluster (February/March 2020) started on a multi-day trip. One player had a sore throat and fever four days before the trip. As he received a negative PCR result, he went on a trip with the club. After two days, two more players developed symptoms, such as abdominal pain and malaise. The symptomatic players were sent home, while other players continued their trip, training, and matches. The players who were sent back home were tested with PCR immediately after return and the results were positive. In the meantime, the vast majority of players developed symptoms. Upon their return, all players were tested, following positive PCR test results.

The fourth cluster probably occurred due to the participation of several players in the traditional New Year’s indoor soccer tournament, which had taken place over several days in a closed sports hall with the presence of the large audience (this was allowed at the time). It was assumed that the players had been infected there since many new cases of COVID-19 among the general population were linked to that tournament. During and after the tournament, the players had their regular training at the club, which raised the possibility of spreading the virus in the incubation period to other players. Given that it was a new virus variant, Omicron VOC, which was more transmissible than earlier variants, the infection spread very fast among the players, and even among those who had previously recovered from COVID-19 (six of them were reinfected). Two vaccine breakthrough infections (infection in vaccinated players) were reported.

In season 1 eight players were asymptomatic and two players in season 2 which is not of statistical significance (χ^2^ = 1.116, *p* = 0.291). The most frequent symptoms in season 1 were fatigue (61.3%), malaise (58.1%), fever (48.4%), headache (45.2%), and anosmia (41.9%), while in season 2021–2022 runny nose/stuffy nose (50%), headache (31.3%), and cough (31.3%) were frequently reported. Fever was notified in only one (6.1%) player. There was no severe disease requiring hospital care or specific COVID-19 treatment (e.g., oxygen, dexamethasone) (Table 2).

There was no report of cardiac or skin symptoms. The most of infected players in season 1 had six or more symptoms (41.9%), while in season 2 majority of infected players had up to two symptoms (68.75%) (Figure 2). A significant difference was found in the average number of reported symptoms per player (5.2 in season 1 vs. 2.1 symptoms in season 2) (*t*-test = 3.24; *p* = 0.002) Cohen’s d = 0.8.

Among infected players in season 1, no significant difference was found in the average number of symptoms per player for those who were infected for the first time vs. those who were reinfected (*t*-test = 1.423, *p* = 0.181) with the mean value of 2.9 for the first time infected, and 2.7 for the reinfected players. During season 1, 67.7% of players had no physical activity for 10 or more days (average 10.3 days), while in season 2, most players (56.3%) had no physical activity for up to 4 days (average 5.1 days) which was found to be a significant difference (χ^2^ = 10.000; *p* = 0.006) (Figure 3a). It was found that the number of symptoms significantly predicted the number of days with no physical activity. The fitted regression model was: y = 0.7634x + 5.3759. The overall regression was statistically significant (R^2^ = 0.3177, F(1, 45) = 20.9523, *p* < 0.001) (Figure 3b).

The infected players’ self-assessed number of days to return to full form was up to 7 days after termination of isolation for 58.1% players in season 1 and for 93.8% in season 2 (χ^2^ =: 6.744; *p* =: 0.034). However, 22.6% players self-reported to be in full form after 15 or more days in season 1 (60 days being the highest value, as reported by two players), while in season 2 it was not reported by any player (Figure 4a). In season 2, reinfected players reported significantly shorter self-assessed absence of full form (mean = 4.3 days) than first time infected players (mean = 6.3 days) (*t*-test = 2,197, *p* = 0.048), Cohen’s d = 1.1. It was found that the number of symptoms significantly predicted self-assessed absence of full form. The fitted regression model was: y = 1.9743x + 2.9564. The overall regression as statistically significant (R^2^ = 0.3966, F (1, 45) = 29.577, *p* < 0.001) (Figure 4b). 

There were 16.1% players (5 out of 31) in season 1 with symptoms after isolation (fatigue, malaise, fever, headache, insomnia) unlike season 2. The presence of fever significantly extended the number of days until return to the self-assessed full form (analysed for both seasons together) (*t*-test = 2422; *p* = 0.027), Cohen’s d = 1.0. The mean self-assessed number of days to return to full form for the group having reported the presence of fever was 19.0, while for the group without the fever the mean was 7.0 days. After discontinuation of isolation, 25 out of 31 infected players (80.60%) in season 1 underwent medical examination and laboratory and cardiac screening according to the decision of the medical team, while in season 2 all infected players were examined. Chest X-rays were performed upon clinical suspicion, in three players in season 1 with no further pathological findings. Cardiac tests (12-lead electrocardiogram, ergometry, heart ultrasound) found no abnormal findings among tested players in both seasons. 

Abnormal laboratory findings were detected when complete blood counts with differential tests, comprehensive metabolic panel, lipid panel and urine test strips were analysed both in season 1 (40% of infected players) and season 2 (87.5%). In season 1, out of 31 infected players, 25 of them underwent laboratory testing. Abnormal findings were reported in 10 players. In season 2, abnormal laboratory findings in 14 out of the 16 infected players were observed. Significantly more players with abnormal laboratory findings were identified in season 2 (χ^2^ = 9.069240; *p* = 0.002) (Figure 5). 

The most common abnormal laboratory findings in season 1 were a higher level of AST (27.6%), CK (24.1%), and ALT (13.8%) while in season 2 a higher level of CK (29.4%), Ca (23.5%), blood platelet, iron, TIBC, and UIBC (17.6% each) were reported. All abnormal findings were close to the referent level. C-reactive protein, D-Dimer, hs- troponin, and NT- proBNP were below the normal level in all infected players in season 2. Just 29% players in season 2 (9 out of 31 players) completed at least a primary vaccination course.

As mentioned in the methods section, MRP parameters were analysed prior to and post COVID-19 infection and the same sample of the games was analysed for both infected and noninfected players. Results of the descriptive statistics are presented in the Table 3.

A repeated measures ANOVA (Table 4) was used to identify differences in the MRP. Main effects showed no differences between groups, however higher values were reported in the post MRP regarding the whole sample for running (F = 5.63, *p* = 0.03), high intensity running (F = 14.02, *p* = 0.01), high speed running (F = 7.99, *p* = 0.01), and sprint (F = 6.38, *p* = 0.02). In addition, there were no significant differences in MRP for the interaction effects.

## 4. Discussion

The results, presented in this paper, raised several important issues. First, the players had significantly more symptoms, longer periods of physical inactivity, longer self-assessed periods to reach the full form, and more frequent symptoms after isolation in season 1 (2020/2021) (time of Wuhan wild strain and Alpha VOC) than in season 2 (2021/2022) (time of Omicron VOC). No cardiac involvement was noticed. Second, despite milder clinical presentation of infection in season 2, the players had significantly more abnormal laboratory findings, although with no clinical significance at the time of the study. Third, low compliance with the preventive measures as well as the low vaccination coverage among players were observed and contributed to the cluster emergences. Fourth, the Omicron variant of the virus did not negatively affect player’s performance, while there was an improvement in the MRP during the post-game period for the sample of all players. 

### 4.1. COVID-19 Epidemiology and Clinical Relevance

Symptoms’ presentation, the severity of the disease, the number of days without physical activity, as well as the time to return to full form after isolation, showed, to a large extent, similarity with other studies. All infected players in both seasons had mild clinical presentation of COVID-19 disease with no desaturation or required hospital care, as shown elsewhere [9,11,12,13,31,32]. However, the proportion of asymptomatic cases in these studies highly varies. This may be explained by different national testing recommendations/club testing protocols, study design, and dominant virus variant. Some sport clubs adopted the strategy of weekly testing to minimise the risk of transmission and avoid the clusters by reducing undetected asymptomatic players which was found to be useful to some extent [32]. In this soccer club, testing was performed only upon indication (after symptom appearance or following occurrence of cluster) and not as a part of a regular testing. 

Moreover, there were significant differences in infection during the two COVID-19 pandemic seasons with different predominant VOCs. We noticed that infection in season 1 showed a more severe clinical course in terms of number of symptoms (6.2 vs. 2.1 in season 2), as well as more days with no physical activity (on average 11.2 vs. 4.8), and a longer period to return to full form (up to 7 days for 58.1% vs. 93.8%). In addition, 25% of the players reported the presence of symptoms after isolation (fatigue, malaise, cough, headache, insomnia) which were not reported in any player in season 2. The reason for these results may be emergence of the new virus in the population with no previous immunity. 

According to the current national guidelines, all positive players were advised to isolate, regardless of symptom presence. A considerable number of players in season 1 (22.6%) reported 15 or more days with no physical activity. During the prescribed isolation, the players did not participate in official training, and their physical activity in the place of isolation depended on their health status. It was supervised by club coaches. Based on the practices presented in other research [17], it would be useful to plan and motivate the players to workout at home during isolation in order to minimise the detraining effect when returning to form. Home trainings have to be planned and structured individually by coaches and strength-conditioning professionals in collaboration with the club medical team with respect to the health condition of each player. In addition, it was presented that the mental health of the players may be improved if coaches are trained to communicate correctly during and with regards to the crisis [33]. This approach may be implemented in any similar situation in which athletes stay at home or away from the club facilities.

The cardiological examinations did not indicate any pathological disorders in all positive players from both seasons. In addition, thromboinflammatory biomarkers and biomarkers of heart failure were negative in all infected players in season 2. The medical team followed protocol proposed by Elliott et al. and Wilson and al. since no official recommendations existed [15,16]. 

A crucial question for professional athletes is the decision regarding the right time to safely return to training and matches, with particular concern towards cardiac events and post-rest injuries. The recommendations and protocols have been evolving during the pandemic and a growing body of evidence has emerged. Such evidence started from the assessment of detailed cardiac screening (ECG, troponin level, heart ultrasound) after infection for all athletes [34]. Successively, it was proposed that cardiac involvement should be examined only in those players with severe and worsening cardiac symptoms [35].

The newest recommendations of American College of Cardiology revised a previous statement and recommended that in case of low suspicion for cardiac involvement (no chest pain/pressure, dyspnea, palpitations, and syncope), there is no need for cardiac testing, which is in line with this study [36]. That screening should be performed with increased clinical suspicion of cardiac involvement or persistent COVID-19 symptoms [16,17,37,38,39]. 

Despite the milder clinical presentation in season 2, there were significantly more abnormal laboratory findings than in season 1, although with no clinical significance. These findings cannot be explained with certainty because they cannot be compared with previous finding as they were not available. Moreover, soccer club physicians continued to follow the players during the three months after the end of the 2021–2022 season and concluded that there were no current health consequences. Due to the lack of data related to these results, the belief is that additional research with more with more participants included should be conducted in order to draw appropriate conclusions.

Most of the infected players could be linked to transmission chains. From the first cluster notified in this club, the epidemiological team proposed a detailed protocol aiming to interrupt and prevent further transmission which was updated several times during the pandemic. Three clusters in season 1 were caused by the infection spreading in the susceptible population due to the new virus emergence and lack of the adherence with the control measures. In that time, only non-pharmaceutical measures (maintaining physical distance, masks, hand hygiene, disinfection) were available. Although adherence to these control measures was not monitored in this study, the authors of this study were directly involved in outbreak management and were aware that players and staff did not follow proposed control measures. The fourth cluster may be explained by the rapid spread of highly transmissible virus variants and low compliance with preventive measures including low vaccination coverage, although vaccine breakthrough infections were known to be present with Omicron VOC.

As Croatia has performed routine genomic monitoring of SARS-CoV-2 in collaboration with the ECDC, the virus variants causing the appropriate cluster in this soccer club may be identified. The first two clusters of COVID-19 in season 1 were probably caused by the Wuhan wild strain of the SARS-CoV-2, while the third cluster was caused by the Alpha VOC, dominant in the population in February 2021. Interestingly, this virus variant was first identified in Croatia in this soccer club as a part of intensive epidemiological surveillance of the authors of this study. A similar situation occurred in season 2. The sudden spread of COVID-19 among the club’s players and the large number of infected players, six of whom were reinfected, raised the suspicion of Omicron VOC circulation, dominant at that time in Europe. At the time of the fourth cluster observed in season 2 in December 2021/January 2022, Omicron rapidly replaced the Delta variant as the dominant variant, and by week 2 of 2022 Delta constituted less than 15% of all Croatian positive samples [22]. A sharp increase in the number of cases was observed during Omicron dominance, because this variant evades the existing immunity and is inherently more transmissible than previous variants [22].

The Omicron variant has been the most highly mutated strain so far, with more than 50 mutations accumulated throughout the genome. These mutations can increase infectivity and immune escape of the Omicron variant compared with the early wild-type strain and the other four VOCs [40]. In this particular soccer club, the first documented infection with Omicron VOC in Split, Dalmatia County (the biggest county in Croatia) was identified. In addition to the high infectiousness of Omicron VOC, the low implementation of both pharmaceutical and non-pharmaceutical control measures (despite recommendation of medical teams and current national recommendations) contributed to the high number of infected players. These clusters are similar to the corresponding waves notified nationwide in the same period of time, as well as pandemic waves that were recorded simultaneously across Europe [41,42]. It is reasonable to assume that the disease incidence in the community would impact the burden of the disease among the players and should thus be accounted for as presented in a study by Papagiannis [32]. 

Current evidence suggests that SARS-CoV-2 has been transmitted mainly through direct contact related to large respiratory droplets and on a lesser extent through droplet nuclei, especially in crowded, enclosed spaces with inadequate ventilation and prolonged exposure. As presented by Schreiber et al., direct contact in professional soccer team is short and not frequent (<1 per player-hour for 88% players, each lasting no longer than 3 s) [43]. One video-based analysis of 50 football matches showed that frontal contacts occurred less than once per player per match with none lasting longer than 3 s, while crowding, which on the average included between two and six players, lasted mostly less than 10 s. The authors concluded that the risk of transmission is low during soccer competitions which is in the line of evidence that outdoor sport activities in general are associated with low risk of COVID-19 transmission [44,45]. In this study, the spreading of infection was mostly due to the indoor transmission (the club backroom and during bus travel) and not likely to be associated with on-field transmission. 

However, the high incidence of COVID-19 infection with four clusters in the studied soccer club in both seasons affected the normal functioning of the club, in terms of absence from participating in matches and regular training. In order to minimise the disruption in team performance and health impact on players at individual and family levels, it is necessary to implement and respect control measures.

During season 1, the vaccine against COVID-19 was either not available or was only available for at-risk groups, which did not include young and healthy athletes. Despite the availability of vaccines with time, especially in season 2, vaccination coverage among this soccer club was not high (9 out of 31 players were provided with at least primary course). Two players were infected despite being vaccinated, which is in accordance with the knowledge of immune escape of the Omicron VOC. Low vaccination coverage among soccer players (29%) in this study is lower than the vaccination coverage in general Croatian population (51% in the similar age groups) [46]. However, this result may be explained by the fact that this group of elite athletes has no comorbidities as an additional factor to vaccination as in general population. Although the abovementioned opinions that on-field transmission risk of SARS-CoV-2 in soccer is very low, it is the authors belief that the vaccination of professional players should be recommended and followed, especially since the infections probably spread indoors. These athletes had a notable loss of training days due to COVID-19. The authors agree with Krzywanski et al. that vaccination against COVID-19 may cause smaller and predictable loss and should be included in prevention policies for athletes [47]. Actually, the American National Football League requires primary and booster vaccination for all staff members who are in direct contact with players. It is interesting that, although the vaccination is not required for players, the vaccination coverage is high (>94% players) [48]. It was shown that vaccination against SARS-CoV-2 appears to be well tolerated and associated with few significant side-effects, which were short-lived and did not affect sport participation [49,50]. Regarding other vaccination of professional players, respectable vaccination coverage was reported. As presented by Signorelly et al., a vaccination coverage with flu vaccine among elite soccer players ranged between 40–50% was reported in Italy [51]. In Greece 87% of the medical teams recommended seasonal influenza vaccine, 62% hepatitis B vaccine, and 50% pneumococcal vaccine [52].

### 4.2. COVID-19 Related Running Performance

The results suggest that the Omicron variant of the virus did not negatively affect player’ performance and that there are two major reasons for such a finding. Firstly, as was mentioned previously, epidemiological examination showed milder clinical presentation in infected players in season 2 comparing with the season 1. In addition, post-COVID-19 examination showed that players subjectively felt better and has less symptoms, and perhaps most importantly, spent less days with no physical activity. This for sure resulted in smaller decline of their endurance capacities during infection and quarantine period.

An additional reason for the lack of a significant decline in running performance results as a consequence of infection can be found in return to play protocols. In particular, in the study from 2021 carried out with the players from the same team, a significant decrease in intensity of game running was found; more specifically, a lower number of high intensity accelerations and decelerations in the infected players was discovered [4]. A major conclusion was that although RTP was well designed and implemented, certain adjustments were needed to further minimise the decline in the mentioned intensive actions [4]. Since some of the authors of both studies (previous and current) are fitness coaches and members of the medical department in the team observed herein, findings from the previous study were systematically applied in practice. In particular, the intensity of the RTP was gradually raised, and the players were involved in football-specific exercises, both at an individual and group level, earlier than those with the first infection. However, it must be mentioned that all the latter was carried out in controlled conditions (i.e., control of heart rate and mechanical load).

Despite the global importance of this topic, there are not many studies that have directly compared the performance of athletes before and after COVID-19 infection. One such study was conducted on high school female soccer players and also detected a decrease in intensity, specifically the distance sprinted during the game, due to infection [53]. Therefore, it can be suggested that control and proper dosage of exercise intensity during RTP are essential for maintaining specific fitness abilities of athletes. Together with previously mentioned milder symptoms, the more accurate RTP protocol almost certainly contributed to the fact that the teams’ players did not experience a decrease in performance as a result of COVID-19 infection during the competitive season 1.

One of the differences found in the observation of match running performance is the general increase of the distance covered in running (15–19.8 km/h), high speed running (19.8–25 km/h), and sprinting (>25 km/h) zones for both the infected and non-infected groups. However, there is a logical explanation for such results considering the specific time of the season when all data were collected. As all infections occurred during late December and the beginning of the January, all games in the baseline measurement were scheduled for late November and December, i.e., at the very end of the first half-season that started on 15 June. On the other hand, all games played during follow-up were during and immediately after the winter preparation period. Consequently, it is reasonable to suppose that increases in mentioned running parameters occur as consequence of higher fitness level after a period of specific training, where one of the primary goals was to increase fitness capacity. Supportively, this is in accordance with previous studies that regularly confirm improvement in aerobic and anaerobic capacities after pre-season training period [54,55]. For medical teams, the decision related to safe return to play is challenging and should be carefully determined, ideally on a case-by-case basis, considering COVID-19 clinical presentation. The return to training must be gradual and supervised by health professionals.

### 4.3. Strengths and Limitations

Many studies of different aspects of COVID-19 and how it impacts sport have been published so far, but to the authors’ best knowledge, this is the first detailed comparison of clinical presentation and epidemiological characteristics as well as MRPs on infected soccer players during two consecutive professional soccer seasons with different dominant SARS-CoV-2 VOC. 

The strengths of this study include a respectable period of observation, although the study was induced and started after the appearance of the first cluster in the club. In addition, the players of the club were continuously and closely monitored for two professional seasons, which is a substantial period of follow-up. In addition, the instruments used to evaluate MRP were previously validated GPS devices that guarantee precise measurements of a player’s movements.

Caution should be taken considering the small sample size. Additionally, all included players were the members of a single team competing in the Croatian league. As a result, these findings may not be generalisable to the general soccer population. However, it was assumed that the health impact regarding symptoms and complications as well as the differences in MRP pre and post COVID-19 infection were not associated to a specific soccer club rather than to the SARS-CoV-2 variants which shared the same emergence and burden in whole Croatia. In addition, the focus on elite soccer players limits the extrapolation of the results to other sport disciplines. This research was focused on a population of adult men, who are likely to be healthier than the general population. Since the symptoms were recorded after a positive test, it may be that such reporting is subject to information. In addition, players may be inclined to give desirable answers due to possible consequences for their playing status in the club. The authors tried to overcome this limitation by carrying out a prompt interview lead by experienced epidemiologist.

This study was performed pre and during the vaccination era. It cannot be excluded that the vaccine had some impact on the clinical presentation and MRP in season 2 since 29% of the players were vaccinated and two of them had a vaccine breakthrough infection. 

Regarding MRP analysis, there is no clear established methodology in selecting a post-infection time frame for the game analysis, so the authors, as in the previous study, took a period of one month for the analysis. In addition, it should be emphasised that several other factors possibly could affect MRP (i.e., opponent, tactics, result, etc.).

## 5. Conclusions

In this study of soccer players, COVID-19 was associated with a mild, self-limited, or asymptomatic infection, not requiring hospital care, regardless of dominant VOC. Regarding cardiac involvement, there were no findings of any cardiac complication. 

The authors further presented transmission chains during several clusters which raised concerns about the lack of adherence to preventive measures, not only during in-club time, but in the private life of the players which affected team performance. Despite limited COVID-19 vaccine effectiveness, especially during Omicron domination, low-vaccination coverage among players may have contributed to the time loss following COVID-19 infection and consequential impact on matches and training. 

Regarding MRP, the results of this research suggest that certain changes to the RTP protocol, made on the basis of knowledge from previous research on the same sample, were correctly applied as they helped athletes to relatively quickly regain their pre-infection levels of physical capacities. This study advances the understanding that an optimally and individually planned RTP protocol is crucial for the MRP of infected players.

This study extends the current literature base by providing evidence on abnormal hematologic and biochemical findings in a group of previously healthy athletes, further presented during the Omicron season, although with no clinical significance at the time of the study.

The results from this study helped us to better understand the effect of different prevailing SARS-CoV-2 VOCs on players’ health and performance. Future research is needed to replicate the findings of abnormal laboratory results and to extend the study, focusing on potential long-term clinical significance. 

## Figures and Tables

**Figure 1 ijerph-20-01950-f001:**
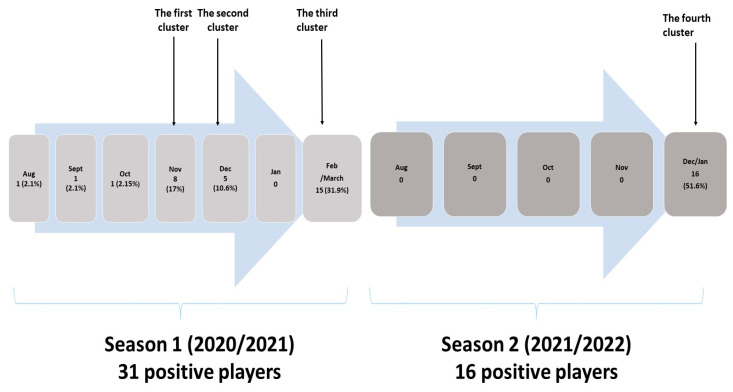
Infected players during two competitive seasons in studied soccer team.

**Figure 2 ijerph-20-01950-f002:**
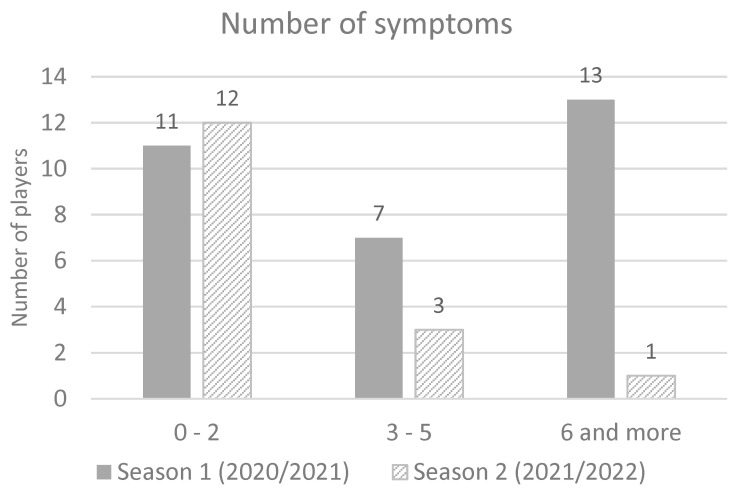
Number of symptoms in season 2020/2021 and season 2021/2022.

**Figure 3 ijerph-20-01950-f003:**
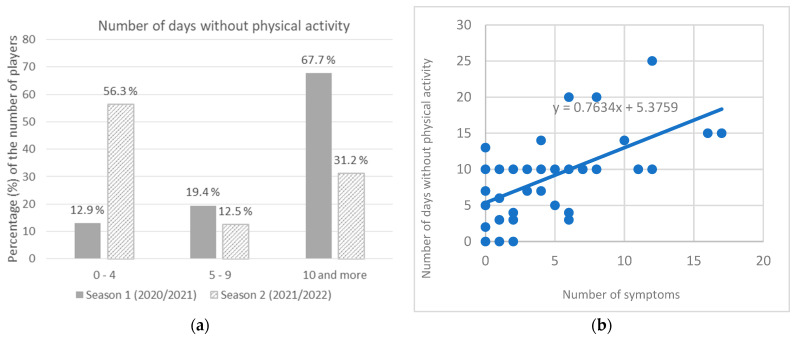
(**a**) Number of days with no physical activity among infected players; (**b**) Correlation of number of symptoms and number of days with no physical activity.

**Figure 4 ijerph-20-01950-f004:**
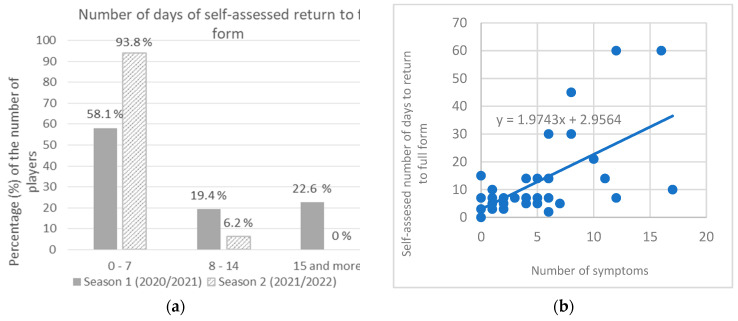
(**a**) Self-assessed number of days to return to full form; (**b**) correlation of number of symptoms and self-assessed number of days to return to full form.

**Figure 5 ijerph-20-01950-f005:**
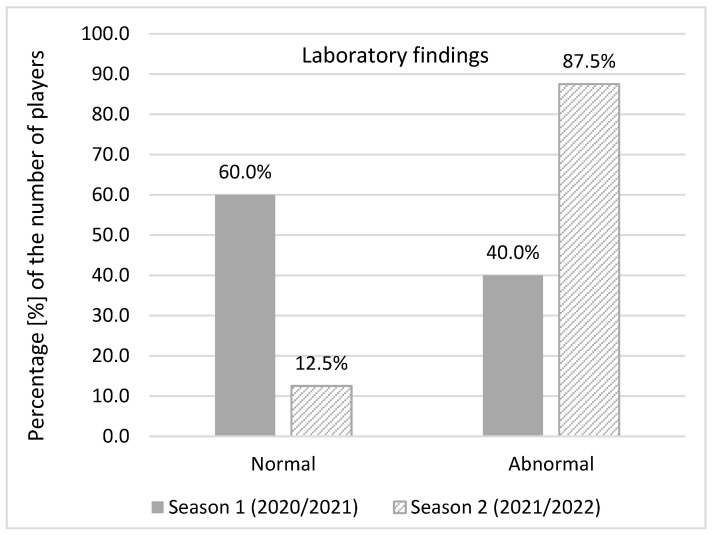
Abnormal laboratory findings.

**Table 1 ijerph-20-01950-t001:** Distribution and characteristics of players.

	Season 1 (2020/2021) No (%)	Season 2 (2021/2022) No (%)	χ^2^
Total number of players	47	31	
Positive test on SARS-CoV-2	31 (66.0%)	16 (51.6%)	χ^2^ =1.604
Age range in years (Median)	18–40 (21)	17–37 (24)	
Previously infected with SARS-CoV-2	1 (2.1%)	15 (48.4%)	χ^2^= 24.516
Re-infected among all with positive test	1 (3.2%)	6 (37.5%)	χ^2^= 9.780
Vaccinated players	0	9 (29%)	

**Table 2 ijerph-20-01950-t002:** COVID-19 symptoms among infected players.

Symptoms	Season 1 (2020/2021) No (%)	Season 2 (2021/2022) No (%)
Asymptomatic	8 (25.8%)	2 (12.5%)
Fever *	15 (48.4%)	1 (6.3%)
Headache	14 (45.2%)	5 (31.3%)
Chills	2 (6.5%)	0
Anosmia	13 (41.9%)	1 (6.3%)
Ageusia	12 (38.7%)	2 (12.5%)
Fatigue	19 (61.3%)	2 (12.5%)
Malaise	18 (58.1%)	2 (12.5%)
Pain in muscles and joints	8 (25.8%)	1 (6.3%)
Cough	9 (29.0%)	5 (31.3%)
Runny nose/congestion	8 (25.8%)	8 (50.0%)
Shortness of breath	6 (19.4%)	0
Sneezing	3 (9.7%)	0
Gastrointestinal symptoms	4 (12.9%)	0
Sore throat	14 (45.2%)	3 (18.8%)
Polyuria	2 (6.5%)	0
Weight loss	4 (12.9%)	1 (6.3%)
Insomnia	4 (12.9%)	0

* more than 37 °C.

**Table 3 ijerph-20-01950-t003:** Descriptive statistics for running parameters.

	Infected		Non Infected	
	Mean	SD	Mean	SD
Total distance PRE	11,083.71	1048.29	10,793.89	1087.88
Low-intensity running PRE	8516.86	949.43	8523.56	415.56
Running PRE	1663.65	282.91	1647.32	550.90
High-intensity running PRE	764.79	233.71	622.84	266.67
High-speed running PRE	618.19	170.38	507.64	212.42
Sprint PRE	145.10	76.24	115.12	63.26
Accelerations PRE	493.83	55.97	503.36	52.93
Decelerations PRE	482.59	57.44	498.17	54.86
High-intensity accelerations PRE	26.76	11.34	23.08	8.50
High-intensity decelerations PRE	39.13	12.56	36.27	7.89
Total distance POST	11,029.33	654.84	11,046.77	718.58
Low-intensity running POST	8380.49	302.26	8502.06	371.54
Running POST	1799.93	296.31	1803.83	430.69
High-intensity running POST	848.46	205.05	740.35	215.46
High-speed running POST	680.45	165.94	595.58	168.10
Sprint POST	168.15	56.88	144.67	59.36
Accelerations POST	495.61	34.81	522.05	40.49
Decelerations POST	494.84	38.17	518.94	38.12
High-intensity accelerations POST	26.85	8.28	27.66	8.53
High-intensity decelerations POST	39.89	10.79	39.30	11.44

**Table 4 ijerph-20-01950-t004:** Repeated measures of match running parameters.

	Main Effects						Interactions		
	Groups			Pre-Post					
	F	*p*	Eta	F	*p*	Eta	F	*p*	Eta
Total distance	0.13	0.72	0.01	0.19	0.67	0.01	0.46	0.51	0.03
Low-intensity running	0.10	0.75	0.01	0.13	0.72	0.01	0.07	0.80	0.00
Running	0.00	0.97	0.00	5.63	0.03	0.27	0.03	0.87	0.00
High-intensity running	1.30	0.27	0.08	14.02	0.00	0.48	0.40	0.54	0.03
High-speed running	1.37	0.26	0.08	7.99	0.01	0.35	0.23	0.64	0.02
Sprint	0.78	0.39	0.05	6.38	0.02	0.30	0.10	0.76	0.01
Accelerations	0.91	0.36	0.06	0.60	0.45	0.04	0.41	0.53	0.03
Decelerations	1.28	0.28	0.08	1.06	0.32	0.07	0.07	0.79	0.00
High-intensity accelerations	0.12	0.73	0.01	1.26	0.28	0.08	1.16	0.30	0.07
High-intensity decelerations	0.11	0.75	0.01	1.96	0.18	0.12	0.70	0.42	0.04

## Data Availability

Data will be provided to all interested parties upon reasonable request.

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
