# Peer review of "How Different Predominant SARS-CoV-2 Variants of Concern Affected Clinical Patterns and Performances of Infected Professional Players during Two Soccer Seasons: An Observational Study from Split, Croatia"

_ijerph, 2023, doi:10.3390/ijerph20031950_

Round 1
Reviewer 1 Report
Title: COVID-19 Among Professional Soccer Players During Two 2 Soccer Seasons with Different Predominant SARS-CoV-2 Vari-3 ants of Concern
1) The abstract is ok. Authors should mention the study location in the abstract as authors only mentioned country name in the abstract. Authors should mention the methods name used to analyse the data here.
2) In the “1. Introduction”, authors only cites one paper in the following lines. They should cite more relevant papers here to reflect more significance:
“For the last three years the world has been changing dramatically due to the COVID-19 pandemic. A variety of control measures, especially lockdowns on different levels, stopped or limited many aspects of social life, including sport. This pandemic profoundly influenced sport as one of the generators of social and economic development, regardless its level: professional, amateurish or recreational. Heavy burden of this pandemic on professional sport was manifested through the restriction of opportunities for training as well as cancellation or postponement of sport events with serious financial loss. It seems to be clear that we cannot avoid the virus for the rest of our lives, but we can minimize its impact, even in sports.”
I recommend the authors to enrich the literature review of the paper. For instance, the authors missed one of early evidence which talked about football players during Covid 19 pandemic.
https://doi.org/10.1002/smi.3059
It will be better if authors add organizations of the paper at the end of introduction section.
3) 2. Materials and Methods:
Authors should be clear regarding the data collection procedures. Authors only mentioned the country name for the study. So, they should mention the study location for more authentication of the study. Which type of data was collected- cross-sectional or longitudinal? Which sampling technique was adopted? How many responses were considered in the study? Authors mentioned that they used structured questionnaire. It will be better if authors mention the survey questionnaire in the “2. Materials and Methods section or in the appendix”. Authors should explain the data analysis in details in this section.
4) “3. Results” is ok.
5) Authors started the conclusion with the main result of the study. But, they should started the conclusion with the main objective of the study. More recent relevant should be put here in relation to the study output that will reflect more significance of the study.
6) Authors should put the implications of the study, limitations and future directions
7) It will be nice if authors check the references with in-text citations. They should also check the tables and figures.
Good-Luck
Author Response
Thank you very much for all your comments and suggestions. We really appreciate your time, energy, goodwill and intention to improve our paper. All your questions have been answered and all your suggestions have been accepted. We hope that the scientific quality of this work has met the high criteria of this respected journal and that it is now suitable for publication. We are willing to make any additional changes if necessary.
1) The abstract is ok. Authors should mention the study location in the abstract as authors only mentioned country name in the abstract. Authors should mention the methods name used to analyse the data here.
Response: Thank you for this suggestion. We added these in the abstract as well as in the title.
2) In the “1. Introduction”, authors only cites one paper in the following lines. They should cite more relevant papers here to reflect more significance:
“For the last three years the world has been changing dramatically due to the COVID-19 pandemic. A variety of control measures, especially lockdowns on different levels, stopped or limited many aspects of social life, including sport. This pandemic profoundly influenced sport as one of the generators of social and economic development, regardless its level: professional, amateurish or recreational. Heavy burden of this pandemic on professional sport was manifested through the restriction of opportunities for training as well as cancellation or postponement of sport events with serious financial loss. It seems to be clear that we cannot avoid the virus for the rest of our lives, but we can minimize its impact, even in sports.”
I recommend the authors to enrich the literature review of the paper. For instance, the authors missed one of early evidence which talked about football players during Covid 19 pandemic.
https://doi.org/10.1002/smi.3059
Reponse: With this comment, you stimulated us to improve our introduction thank you!
We carefully read more literature and we included this and some other studies in this manuscript and in the reference list.
It will be better if authors add organizations of the paper at the end of introduction section.
Response: As suggested, we added, thank you.
3) 2. Materials and Methods:
Authors should be clear regarding the data collection procedures. Authors only mentioned the country name for the study. So, they should mention the study location for more authentication of the study.
Response: Thank you for this observation. We added this clarification in the manuscript.
Which type of data was collected- cross-sectional or longitudinal?
Reponse: Thank you for this question. We assumed this study as two longitudinal studies, in season 1 and season 2, since the players were not the same in both season. We added this clarification in the manuscript: „
This observational study includes two whole populations of professional male soccer players from the first division of one soccer club in Split, the second biggest city in Croatia. This club participates in the first national soccer league along with nine other soccer clubs. One population consisted of all players (N 47) in season 2020/2021 (Season 1) and the other consisted of all players (N 31) in season 2021/2022 (Season 2).“
Which sampling technique was adopted?
Response: In both seasons, all nasopharyngeal and pharyngeal swabs were tested using real-time reverse transcriptase polymerase chain reaction (RT-PCR) for the detection of SARS-CoV-2 at certified laboratories by Croatian Health Insurance Fund. These swabs were collected by medical staff from official laboratories. Test results were available within 24 hours in most cases. According to regulations of the Croatian national soccer federa-tion, there was no obligation to test during the season or before matches. The players were tested following the presence of symptoms and during a cluster investigation when all the team was tested. The testing was initiated by the team physician or epidemiologist and it was not repetitive for non-infected players unless the presence of the symptoms occurred.
We hope that the process of sampling is more clearer now for you and for potential readers, thank you!
How many responses were considered in the study?
Response: Thank you for this question. All players in both seasons were included in this study. There was no refusal of testing after a doctor's recommendation or to MRP measurement. We enclosed this statement in the manuscript as „All players in both seasons were included in the study. There was no refusal of testing after a doctor's recommendation or to MRP measurement.“
Authors mentioned that they used structured questionnaire. It will be better if authors mention the survey questionnaire in the “2. Materials and Methods section or in the appendix”.
Authors should explain the data analysis in details in this section.
Response: We added more details in the manuscript: The epidemiologist interviewed players using standard data collection for COVID-19 case reporting in Croatia in the general population [ref] which was changed over time with additional questions regarding vaccination and persistent symptoms. The in-terview was done immediately after a positive test, and then again one month after testing. Some extra questions were added regarding the number of days to return to full form after the termination of isolation and the number of days with no physical activity, during and after isolation caused by a COVID-19 infection. These questions were pilot tested for clarity and consistency on several participants and modified accordingly. „
We hope that the process of data collection is more clearer now for you and for potential readers.
4) “3. Results” is ok.
Response: Thank you!
5) Authors started the conclusion with the main result of the study. But, they should started the conclusion with the main objective of the study. More recent relevant should be put here n relation to the study output that will reflect more significance of the study.
Response: Thank you for this suggestion. We started our conclusion with main objective and organised this section as you suggested.
6) Authors should put the implications of the study, limitations and future directions
Response: We thank you for this helpful observation. We expand discussion and conclusion with these points as „In this study of soccer players, COVID-19 was associated with a mild, self-limited or asymptomatic infection, not requiring hospital care, regardless dominant VOC. This study extends the current literature base by providing evidence on abnormal he-matologic and biochemical findings in a group of previously healthy athletes, further more presented during the Omicron season, although with no clinical significance at the time of the study. Regarding cardiac involvement, there were no findings of any cardiac complication.
The authors further presented transmission chains during several clusters which raised concerns about the lack of adherence to preventive measures, not only dur-ing in-club time, but in the private life of the players which affected team performance. Despite limited COVID- vaccines effectiveness, especially during Omicron domination, low vaccination coverage among players may have contributed to the time loss following COVID-19 infection and consequential impact on matches and trainings.
Regarding MRP, the results of this research suggest that certain changes to the RTP protocol, made on the basis of knowledge from a previous research on the same sample, were correctly applied as they helped athletes to relatively quickly regain their pre-infection levels of physical capacities. This study advances the understanding that an optimally and individually planned RTP protocol is crucial for the MRP of infect-ed players.
The results from this study helped to better understand the effect of different prevail-ing SARS-CoV-2 VOCs on player's health and their performances. Future research is needed to replicate the findings of abnormal laboratory results and to extend the study focusing on their potential long-term clinical significance.“
7) It will be nice if authors check the references with in-text citations. They should also check the tables and figures.
Response: Thank you for this proposal. We put in-text citation using EndNote and we checked the tables and figures. In addition, the paper was corrected by native speaker.
Good-Luck
Response: Thank you again!
Reviewer 2 Report
Dear authors it is an interesting and well written article, I find it really innovative and useful for suggest monitoring programs for COVID 19 among different sports and players
A general suggestion; please revise spelling of some words
Specific clarifications
Materials and Methods
Professional players with less than 10 year where not included?
Is not clear when the nasal swab was taken. After the each match? In a clinical control? Was it repetitive for NONIF?,
Who takes the sample?
Results
31 players where included; are just from one team (club)? or how many teams were included? there are no more professional players in other teams? if players come from different teams. how they were selected? perhaps this could be clarified on methods
Perhaps to include χ2 on the table 1 could be usefully to a better understanding
Author Response
Dear authors it is an interesting and well written article, I find it really innovative and useful for suggest monitoring programs for COVID 19 among different sports and players
Thank you
Thank you very much for all your comments and suggestions. We really appreciate your time, energy, goodwill and intention to improve our paper. All your questions have been answered and all your suggestions have been accepted. We hope that the scientific quality of this work has met the high criteria of this respected journal and that it is now suitable for publication. We are willing to make any additional changes if necessary.
A general suggestion; please revise spelling of some words
Response: Thank you for this suggestion. The paper was corrected by native speaker.
Specific clarifications
Materials and Methods
Professional players with less than 10 year where not included?
Response: Thank you for opportunity to clarify our inclusion criteria. All players in both seasons were included in this study. However, for the purpose of this study, we included MRP measurement of players who have had professional status on team during both season and who have actively participated in soccer for at leat 10 years. We added this distinction in the paper.
Is not clear when the nasal swab was taken. After the each match? In a clinical control? Was it repetitive for NONIF?
Who takes the sample?
Response: In both seasons, all nasopharyngeal and pharyngeal swabs were tested using real-time reverse transcriptase polymerase chain reaction (RT-PCR) for the detection of SARS-CoV-2 at certified laboratories by Croatian Health Insurance Fund. These swabs were collected by medical staff from official laboratories. Test results were available within 24 hours in most cases. According to regulations of the Croatian national soccer federa-tion, there was no obligation to test during the season or before matches. The players were tested following the presence of symptoms and during a cluster investigation when all the team was tested. The testing was initiated by the team physician or epidemiologist and it was not repetitive for non-infected players unless the presence of the symptoms occurred.
We hope that the process of sampling is more clearer now for you and for potential readers, thank you!
Results
31 players where included; are just from one team (club)? or how many teams were included? there are no more professional players in other teams? if players come from different teams. how they were selected? perhaps this could be clarified on methods
Response: This observational study includes two whole populations of professional male soccer players from the first division of one soccer club in Split, the second biggest city in Croatia. This club participates in the first national soccer league along with nine other soccer clubs.
One population consisted of all players (N 47) in season 2020/2021 (Season 1) and the other consisted of all players (N 31) in season 2021/2022 (Season 2).
We added this clarification in the manuscript, thank you for this suggestion.
Perhaps to include χ2 on the table 1 could be usefully to a better understanding
Response: We include χ2 on the table 1. Thank you again!